# Evaluating Large Language Models as Expert Annotators

**Yu-Min Tseng**$^{\alpha\beta\dagger}$   **Wei-Lin Chen**$^{\gamma}$   **Chung-Chi Chen**$^{\delta}$   **Hsin-Hsi Chen**$^{\alpha\pi}$

$^{\alpha}$National Taiwan University   $^{\beta}$Virginia Tech   $^{\gamma}$University of Virginia
$^{\delta}$AIST, Japan   $^{\pi}$AINTU, Taiwan
ymtseng@vt.edu, wlchen@virginia.edu, c.c.chen@acm.org, hhchen@ntu.edu.tw

## Abstract

Textual data annotation, the process of labeling or tagging text with relevant information, is typically costly, time-consuming, and labor-intensive. While large language models (LLMs) have demonstrated their potential as direct alternatives to human annotators for general domains natural language processing (NLP) tasks, their effectiveness on annotation tasks in domains requiring expert knowledge remains underexplored. In this paper, we investigate: whether top-performing LLMs, which might be perceived as having expert-level proficiency in academic and professional benchmarks, can serve as direct alternatives to human expert annotators? To this end, we evaluate both individual LLMs and multi-agent approaches across three highly specialized domains: finance, biomedicine, and law. Specifically, we propose a multi-agent discussion framework to simulate a group of human annotators, where LLMs are tasked to engage in discussions by considering others' annotations and justifications before finalizing their labels. Additionally, we incorporate reasoning models (*e.g.*, o3-mini) to enable a more comprehensive comparison. Our empirical results reveal that: *(1)* Individual LLMs equipped with inference-time techniques (*e.g.*, chain-of-thought (CoT), self-consistency) show only marginal or even negative performance gains, contrary to prior literature suggesting their broad effectiveness. *(2)* Overall, reasoning models do not demonstrate statistically significant improvements over non-reasoning models in most settings. This suggests that extended long CoT provides relatively limited benefits for data annotation in specialized domains. *(3)* Certain model behaviors emerge in the multi-agent discussion environment. For instance, Claude 3.7 Sonnet with thinking rarely changes its initial annotations, even when other agents provide correct annotations or valid reasoning.[1]

## 1 Introduction

Textual Data annotation refers to the task of labeling or tagging text with relevant information (Tan et al., 2024). For example, adding topical keywords to social media contents. Typically, this process is carried out by crowd-sourced workers (*e.g.*, MTurkers) or specialized annotators (*e.g.*, researchers), depending on the tasks, to ensure high-quality annotations. However, the annotating procedures are often costly, time-consuming, and labor-intensive, particularly for tasks that require domain expertise.

With the rise of large language models (LLMs), a series of works have explored their potential as an attractive alternative to human annotators (Ding et al., 2023; Zhang et al., 2023; Choi et al., 2024; He et al., 2023). Empirical results suggest that, in certain scenarios, LLMs such as ChatGPT and GPT-3.5 even outperform master-level MTurk workers, with substantially lower per-annotation cost (Gilardi et al., 2023; Alizadeh et al., 2023; Bansal & Sharma, 2023; Zhu et al., 2023). However, existing studies mainly focus on general-domain NLP tasks (*e.g.*,

---

[†]Work was done at National Taiwan University.
[1]https://github.com/ymntseng/llm-expert-annotators

sentiment classification, word-sense disambiguation). The extent to which LLMs as data annotators perform in domains requiring expert knowledge remains unexplored.

On the other hand, LLMs have exhibited striking performance in a variety of benchmarks, both professional and academic (Jin et al., 2019; Hendrycks et al., 2020; Chen et al., 2021; Rein et al., 2023; Achiam et al., 2023). Leveraging the abundant domain-specific knowledge encoded in the parameters, LLMs could pass exams that require expert-level abilities (Choi et al., 2021; Singhal et al., 2023a; Callanan et al., 2023; Singhal et al., 2023b; Katz et al., 2024). These findings prompt our research question: Can performant LLMs, which might be perceived as having expert-level proficiency in academic and professional benchmarks, serve as direct alternatives to human expert annotators? We refer to this setting as *LLMs-as-Expert-Annotators*.

To investigate the question, we examine LLMs on three specialized domains: finance, law, and biomedicine. Specifically, we carefully select five existing datasets that *(i)* provide fully-detailed annotation guidelines and *(ii)* are manually labelled by domain experts. We format the annotation task, the guideline, and unlabelled data instances as instructional inputs to the models, and evaluate their annotation results against ground truth labeled by human experts. Toward a more comprehensive evaluation, we employ a variety of inference-time techniques that leverage additional compute to elicit the capabilities of LLMs, using both individual LLM and multi-agent (*i.e.*, multiple LLMs) approaches. Furthermore, inspired by how human annotators reach consensus, we propose a multi-agent annotation framework that allows LLMs to collaboratively generate annotations through discussion.

In sum, our main contributions includes:

- We present one of the first systematic evaluations of LLMs-as-Expert-Annotators and investigate their inability to perform annotation tasks that require specialized domain knowledge.
- We find that, for individual LLMs *(1)* equipping with inference-time techniques demonstrate only marginal or even negative performance gains, contrary to prior literature suggesting their broad effectiveness; *(2)* reasoning models do not exhibit statistically significant improvements over non-reasoning models in most settings.
- We propose a multi-agent discussion framework that enables multiple LLMs to reach stronger consensus, leading to improved performance over individual LLMs. In addition, we conduct a fine-grained analysis of the results and identify specific model behaviors that emerge during the multi-agent discussion process.

Given the high-stakes nature of expert-level annotation in fields such as medicine and finance, our findings highlight a notable gap between current LLMs and human experts, underscoring the need for further advancements before LLMs can be reliably deployed as expert annotators.

## 2 Experimental Setup

### 2.1 Datasets

We evaluate five datasets across three specialized domains: finance, law, and biomedicine. (Task descriptions, dataset statistics, and annotation guidelines are provided in Appendix A and B.) All datasets are multiple-choice tasks. Due to limited resources, we sample 200 instances per dataset, totaling 1000 instances. To ensure data quality and a fair comparison, we have checked these datasets *(1)* provide fully documented annotation guidelines and *(2)* explicitly state that annotation were labeled by human experts and reach consensus.

### 2.2 Models

We experiment with 6 of the most performant, publicly-available language models, including 4 non-reasoning models and 2 reasoning models. The non-reasoning models are `Gemini-1.5-Pro` (Reid et al., 2024), `Gemini-2.0-Flash` (Google, 2024),

`Claude-3-Opus` (Anthropic, 2024), and `GPT-4o` (OpenAI, 2024). The reasoning models are `Claude-3.7-Sonnet` (Anthropic, 2025) with thinking and `o3-mini` (OpenAI, 2025) with medium reasoning effort. For models with a temperature parameter, we set it to 0.0 unless otherwise specified in the method settings.

## 2.3 Evaluation

We assess LLMs-as-expert-annotators by comparing their accuracy against ground-truth labels provided by human expert annotators.

Unlike prior studies that evaluate LLMs on general-domain datasets, we do not compare their performance with crowdworker or non-expert human annotations. Our investigation targets datasets that require specialized domain expertise, which crowdworkers might not be able to provide satisfactory annotations. Further recruiting new human annotators for this study could result in an unfair comparison, as the original dataset annotations were produced by highly selective, qualified experts. Furthermore, using gold-standard annotations provides a more suitable and reproducible test bed for future works to compare the results directly.

## 3 Individual LLMs as Expert Annotators

In this section, we adopt vanilla prompting along with three inference-time techniques: CoT, self-refine, and self-consistency. By leveraging additional inference-time compute, we explore whether individual LLMs can serve as a direct alternative to expert data annotators.

We employ a uniform prompt template that is easily generalizable across all models and tasks. This standardization of prompt phrasing ensures that the only sources of variation in our results are: *(i)* the annotation guideline and *(ii)* the instance to be labeled. We provide all prompt templates in Appendix C.

### 3.1 Methods

**Vanilla**   The vanilla method refers to standard direct-answer prompting, where instructional input consists of the annotation task, guideline, and the instance are given to the LLMs. LLMs are tasked to conduct annotation as a domain expert of relevant fields. The vanilla prompt also serves as the base of other sophisticated approaches (described below).

**CoT**   CoT improves LLMs' complex reasoning ability significantly (Wei et al., 2022). Specifically, we employ zero-shot CoT (Kojima et al., 2022), where a trigger phrase "*Let's think step by step*" augments the prompt to elicit reasoning chain from LLMs and leads to a more accurate answer.

**Self-Consistency**   Self-consistency (Wang et al., 2022) improves upon CoT via a sample-and-marginalize decoding procedure, which selects the most consistent answer rather than the greedily decoded one. Concretely, we sample 5 diverse reasoning paths with temperature 0.7, and take the majority vote to determine the final answer.

**Self-Refine**   Self-refine (Madaan et al., 2024) method includes three steps: generate, review, and refine. An LLM first generates an initial answer (*i.e.*, draft). Then, the model review its draft and provide feedback. Lastly, the LLM refine the draft by incorporating its feedback, and outputs an improved answer. The same LLM is used in all steps.

### 3.2 Inference-Time Techniques Could Undermine LLMs-as-Expert-Annotators

Our results in Table 1 suggest that models struggle to effectively and consistently leverage inference-time techniques, often experiencing performance declines.

Across all models, the application of CoT generally leads to lower accuracy. For instance, Claude 3 Opus exhibits an average accuracy drop of 1.6%. Even when minor improvements

| Model / Method | Finance | | Law | | Biomedicine | Avg. |
| --- | --- | --- | --- | --- | --- | --- |
| | REFinD | FOMC | CUAD | FoDS | CODA-19 | |
| *Claude 3 Opus* | 64.0 | 63.0 | 83.5 | 47.0 | 63.5 | 64.2 |
| *w/* CoT | 62.0 (↓2.0) | 64.5 (↑1.5) | 80.5 (↓3.0) | 43.0 (↓4.0) | 63.0 (↓0.5) | 62.6 (↓1.6) |
| *Gemini 1.5 Pro* | 63.5 | 66.5 | 84.0 | 43.0 | 69.5 | 65.3 |
| *w/* CoT | 59.0 (↓4.5) | 68.0 (↑1.5) | 82.0 (↓2.0) | 37.5 (↓5.5) | 71.5 (↑2.0) | 63.6 (↓1.7) |
| *Gemini 2.0 Flash* | 59.0 | **71.5** | **86.5** | 47.0 | **79.5** | 68.7 |
| *w/* CoT | 62.5 (↑3.5) | 68.5 (↓3.0) | 82.5 (↓4.0) | 45.5 (↓1.5) | 77.5 (↓2.0) | 67.3 (↓1.4) |
| *w/* self-refine | 64.5 (↑5.5) | 68.5 (↓3.0) | 85.0 (↓1.5) | **47.5** (↑0.5) | 76.5 (↓3.0) | 68.4 (↓0.3) |
| *w/* self-consistency | 65.0 (↑6.0) | 70.0 (↓1.5) | 83.5 (↓3.0) | 46.5 (↓0.5) | **79.5** (−0.0) | **68.9** (↑0.2) |
| *GPT-4o* | 67.5 | 68.5 | 84.5 | 44.5 | 74.0 | 67.8 |
| *w/* CoT | 67.0 (↓0.5) | 69.5 (↑1.0) | 84.0 (↓0.5) | 44.0 (↓0.5) | 72.5 (↓1.5) | 67.4 (↓0.4) |
| *w/* self-refine | 66.5 (↓1.0) | 67.0 (↓1.5) | 81.5 (↓3.0) | 45.0 (↑0.5) | 72.0 (↓2.0) | 66.4 (↓1.4) |
| *w/* self-consistency | **69.5** (↑2.0) | 69.5 (↑1.0) | 83.5 (↓1.0) | 46.0 (↑1.5) | 74.0 (−0.0) | 68.5 (↑0.7) |

Table 1: Accuracy of instruction-tuned LLMs on expert annotation tasks. Text in **bold** indicates the highest accuracy for each dataset.

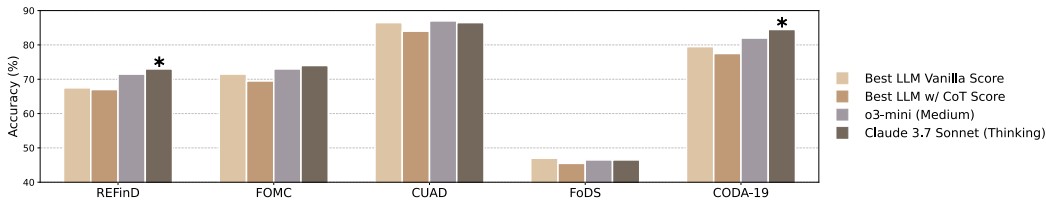

Figure 1: Accuracy comparison between reasoning models and non-reasoning models. An asterisk (*) indicates the reasoning model is statistically significant with p-value $< 0.05$ than the best non-reasoning models with CoT method.

are observed in specific datasets, they are inconsistent and fail to establish a reliable trend of enhancement. Similarly, self-refine and self-consistency show unstable effects. While some cases exhibit slight gains, most results reflect a negative impact, such as GPT-4o experiences an average 1.4% decrease when equipped with self-refine method.

This overall performance decline suggests that these inference-time techniques, which have demonstrated significant performance gains in prior literature, may not be well-suited for the LLMs-as-expert-annotators setting. We speculate that inference-time methods may fail to consistently enhance performance due to fundamental limitations in models' ability to understand complex domain-specific contexts. Specifically, models might not accurately interpret specialized annotation guidelines and input instances, thereby failing to capitalize on the additional inference-time compute, or even degrading performance due to misinterpretation. Therefore, instruction-tuned models appear to struggle with applying these strategies effectively, highlighting a critical limitation in their ability to perform data annotation tasks in specialized domains.

### 3.3 Reasoning Models Outperform Non-Reasoning Models Marginally

As shown in Figure 1, we compare reasoning models (*i.e.*, o3-mini with medium reasoning effort and Claude 3.7 Sonnet with thinking) against the best-performing non-reasoning models across five datasets.

While reasoning models tend to achieve a slightly higher accuracy, the differences are relatively small in many cases. We apply McNemar's test (McNemar, 1947) to assess the statistical significance of their performance differences. The accuracy and corresponding p-values are provided in Appendix D. Across four comparison settings (*i.e.*, each of the two reasoning models versus each of the two non-reasoning models), statistical significance is

observed in only one setting – when comparing Claude 3.7 Sonnet with thinking against the best non-reasoning models with CoT – and in two out of five datasets. These results suggest that, despite their enhanced long CoT inference capabilities, current reasoning models do not yet offer a substantial advantage over non-reasoning models in the LLM-as-expert-annotators setting.

# 4 Multi-Agent Discussion Framework

The multi-agent framework, where multiple LLMs communicate with each other to solve tasks in a collaborative manner, has become a prevalent research direction (Liang et al., 2023; Du et al., 2023; Chen et al., 2023; Tseng et al., 2024b). This approach leverages the collective power of multiple models, enabling them to exchange insights, verify conclusions, and reduce individual biases, ultimately enhancing task performance and decision quality.

In the context of data annotation, a common challenge is the disagreement among multiple annotators, where differing interpretations lead to inconsistent labels. In human annotation workflows, such discrepancies are often resolved through peer discussions, where annotators deliberate over ambiguous cases to reach a consensus. Inspired by this collaborative resolution process, we design a multi-agent annotation framework that incorporates a discussion mechanism. By enabling LLM agents to engage in communication, the framework simulates the consensus-building process of human annotators to assess whether this approach results in more accurate and reliable annotations.

## 4.1 Proposed Framework

---

**Algorithm 1** Multi-Agent Discussion Framework Algorithm

---

**Require:** Set of LLMs $\mathcal{M} = \{M_1, M_2, ..., M_n\}$
**Require:** Annotation task $T$, Annotation guideline $G$, Instance $I$
**Require:** Maximum discussion rounds $R_{\max}$
**Ensure:** Final annotation $\hat{y}$
1: **Initialization:** Set discussion round counter $r \leftarrow 0$
2: **Step 1: Generate Initial Annotation**
3: **for** each $M_i \in \mathcal{M}$ **do**
4:     $\hat{y}_i \leftarrow M_i(T, G, I)$
5: **end for**
6: **while** $r < R_{\max}$ **do**
7:     **Step 2: Check Consensus**
8:     **if** all annotations $\hat{y}_1, \hat{y}_2, ..., \hat{y}_n$ are identical **then**
9:         **return** Final annotation $\hat{y} = \hat{y}_1 = \hat{y}_2 = ... = \hat{y}_n$
10:     **else**
11:         **Step 3: Discuss and Re-Annotate**
12:         Compile all $\hat{y}_i$ and reasoning into Discussion History $D_r$
13:         **for** each $M_i \in \mathcal{M}$ **do**
14:             Generate revised annotation: $\hat{y}_i^{(r+1)} \leftarrow M_i(T, G, I, D_r)$
15:         **end for**
16:         $r \leftarrow r + 1$
17:     **end if**
18: **end while**
19: **Step 4: Majority Vote (if no consensus reached)**
20: $\hat{y} = Majority Vote(\hat{y}_1, \hat{y}_2, ..., \hat{y}_n)$
21: **return** $\hat{y}$

---

Our proposed multi-agent discussion framework involves four steps: *(1)* Generate initial annotations, *(2)* Check consensus, *(3)* Discuss and re-annotate, and *(4)* Majority vote if no consensus reach, as illustrated in pseudo algorithm 1. Initially, each agent generates its own annotation through CoT prompting given the same annotation task, guideline, and instance. Next, we check for consensus (*i.e.*, if all annotations are the same labels). If consensus

is achieved, the instance is successfully annotated and the process completes. If not, we compile all agents' reasoning and labels into a "*Discussion History*". Agents then re-annotate using the same input and discussion history. This check-discuss-re-annotate cycle continues until consensus is achieved or the maximum number of discussion round is reached. We provide the prompt templates of our proposed framework in Appendix C.

In our experiment, we set the maximum number of discussion rounds to 2 based on empirical observations and practical considerations: *(i)* We find that nearly all instances reach consensus within 2 rounds. In fact, fewer than 10 instances fail to reach agreement by the end of the second round. *(ii)* For the rare cases that do not reach consensus after 2 rounds, we observe a common pattern – all three agents resist changing their annotations in at least one round, resulting in no progress.

Through the average performance of individual reasoning models and non-reasoning models, we select three representative model pairings for the multi-agent framework:

- Claude 3.7 Sonnet with thinking, o3-mini, GPT-4o (2 reasoning models)

- Claude 3.7 Sonnet with thinking, GPT-4o, Gemini 2.0 Flash (1 reasoning models)

- GPT-4o, Gemini 2.0 Flash, Gemini 1.5 Pro (0 reasoning models)

To enhance the majority vote accuracy, we set the multi-agent group size (i.e., number of LLMs) to 3, ensuring a more robust and reliable consensus outcome.

## 4.2 Multi-Agent Discussion Leads to Better Agreement and Performance

The multi-agent discussion framework demonstrates its effectiveness by enhancing both accuracy and inter-annotator agreement across all five datasets, as shown in Figure 2.

For performance (top row of the figure), the accuracy of both the discussion framework and each individual agent increases as the discussion rounds progress, demonstrating that collaborative interactions among LLMs lead to more accurate annotations. As the agents iteratively refine their reasoning through communication, the overall performance improves consistently.

Additionally, we calculate the Fleiss' Kappa (Fleiss, 1971) agreement score, which measures the consistency among annotators. As shown in the bottom row of the figure, the agreement score steadily increases with each round, indicating that multi-agent discussions not only enhance accuracy but also promote greater consensus among the models. The final round achieves near-perfect agreement across all datasets, highlighting the robustness and reliability of the multi-agent framework in resolving ambiguous or conflicting cases.

## 4.3 The Discussion Framework Falls Short of Its Upper Bound

For our multi-agent discussion framework, we approximate the upper bound by assuming that agents cannot suddenly generate the correct annotation in the middle of the discussion. The only way to arrive at a correct annotation is if it was present in at least one of the initial agent predictions, and agents are able to identify it and reach consensus through discussion. On the other hand, if the correct annotation is absent from the initial predictions of all three agents, we treat it as an irrecoverable instance. In these cases, regardless of the quality or depth of the discussion, the agents are unable to produce the correct annotation, as it was never part of the initial pool of responses.

As shown in Figure 3, settings containing reasoning models outperform settings containing all three non-reasoning models for multi-agent discussion framework. While the framework consistently outperforms the CoT majority vote (MV) based on the initial annotations of the three agents, it does not always reach its upper-bound performance. The dotted lines reveal a persistent gap ($\Delta$) between the discussion framework and its upper bound.

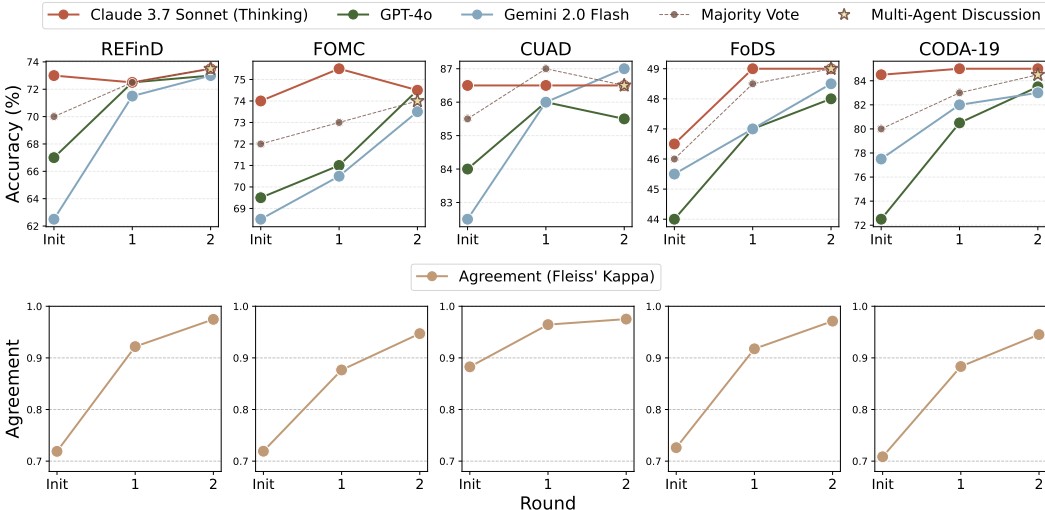

Figure 2: Accuracy and agreement improvements through multi-agent discussion across five datasets (1 reasoning model setting). The top row shows the accuracy progression of each agent and the whole discussion framework. The bottom row displays the Fleiss' Kappa agreement scores, indicating improved consensus among the models.

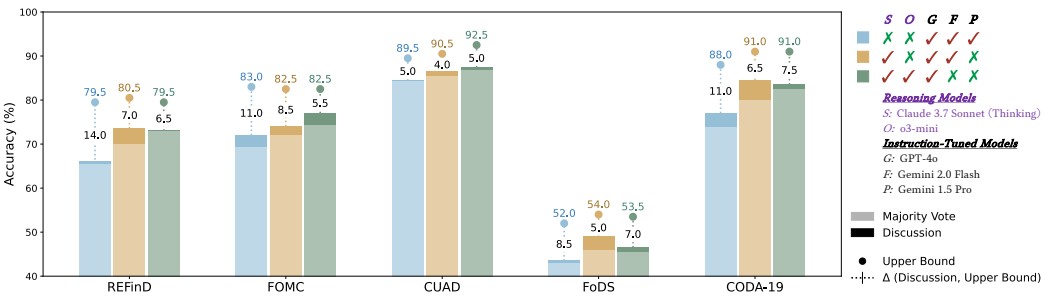

Figure 3: Comparison between multi-agent CoT MV and discussion framework for three model pairing. The dotted lines with numbers indicate the gap (Δ) between the discussion framework and its upper bound.

## 4.4 Emerging Model Behaviors in the Discussion Process

To understand why the discussion framework underperforms relative to its upper bound, we analyze the underlying model behaviors. Since these behaviors are consistent across datasets and settings, we use the FoDS dataset with the 1 reasoning model setting as a representative example, as illustrated in Figure 4. The figure shows the progression of annotations across two rounds (R1 and R2), visualizing how models refine or retain their predictions during the discussion process.

We observe that the reasoning model (Claude 3.7 Sonnet with thinking) rarely changes its initial annotations, regardless of their correctness. While maintaining correct annotations is desirable, this strong self-consistency on incorrect ones limits the potential for collaborative refinement. By contrast, Gemini 2.0 Flash and GPT-4o exhibit greater responsiveness to peer reasoning and a higher willingness to revise annotations. However, their revisions are not reliably targeted toward correcting only incorrect labels. These behavioral tendencies help explain why our multi-agent framework improves over individual models, yet still falls short of the upper bound: collaborative gains are constrained by both strong self-consistency and imprecise revision behavior.

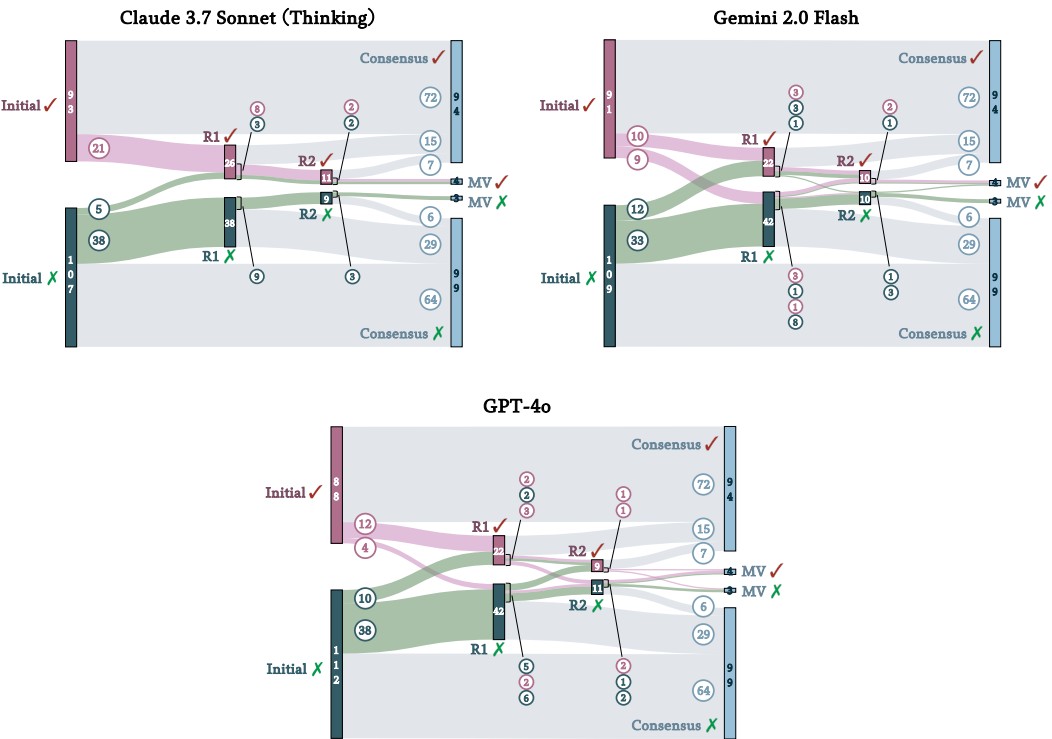

Figure 4: Model behaviors within multi-agent discussion (1 reasoning model setting) on FoDS dataset. The 10 dark-colored nodes represent whether the current annotation is correct at different stages: ▇ dark red (Initial ✓, R1 ✓, R2 ✓), ▇ dark green (Initial ✗, R1 ✗, R2 ✗), and ▇ dark blue (both Consensus ✓/✗ and MV ✓/✗). The 3 light-colored backgrounds indicate how models changed or retained their annotation during discussion: ▇ light red (initial annotation is correct), ▇ light green (initial annotation is wrong), and ▇ light blue (all agents reach a consensus). The consensus (rightmost column) reflects the final agreed-upon annotation, which benefits from the collective refinement process. The numbers in the circles indicate the exact instance counts. Note that the number of instances shown in blue (both dark and light shades) is the same across all three models.

We hypothesize two explanations for the strong self-consistency observed in reasoning models: *(i)* Overconfidence. Strong reasoning models may exhibit higher confidence in their initial annotations, making them less likely to update even when faced with correct annotations from other models. *(ii)* Greater persuasiveness. As shown in prior work (Bozdag et al., 2025; Durmus et al., 2024), prompts that encourage logical reasoning can yield more persuasive arguments. Consequently, reasoning models may generate outputs that are not only logically coherent but plausible, leading others to align with them, even when it's incorrect.

This observation highlights the importance of careful design in future multi-agent systems. Whether this strong self-consistency effect is ultimately beneficial or detrimental remains an open question. Should the effect generalize and contribute to unintended consequences, it would warrant a closer look into the role of reasoning models in collaborative LLM-as-expert-annotator systems, with consideration for appropriate safeguards.

## 5 Related Work

The growing interest in LLMs as automated annotators has led to a surge of studies demonstrating their utility in diverse NLP annotation tasks (Chiang & Lee, 2023; Vu et al., 2024;

Ding et al., 2023; Choi et al., 2024). Early investigations evaluate models like GPT-3.5 and ChatGPT on general-domain tasks reporting performance competitive with or even exceeding that of trained crowdworkers (Gilardi et al., 2023; Alizadeh et al., 2023; Bansal & Sharma, 2023; Zhu et al., 2023). Methods such as active learning (Zhang et al., 2023) and reflection (He et al., 2023) have shown promising results by leveraging the capabilities of LLMs for data annotation tasks. However, these works primarily focus on general-domain tasks, leaving a gap in the exploration of domain-specific applications.

Another emerging trend involves using LLMs in collaborative or multi-agent settings (Guo et al., 2024; Xi et al., 2025). Prior work has explored consensus-building through debate (Du et al., 2023), iterative discussion (Chen et al., 2023), and role-playing simulations (Liang et al., 2023), largely targeting reasoning tasks or factual correctness. However, the application of such frameworks to emulate expert annotation group workflows remains underexplored.

Overall, we build upon previous research (Tseng et al., 2024a) by incorporating reasoning models, and extend the current literature by empirically evaluating both individual and multi-agent LLMs in expert annotation scenarios across finance, law, and biomedicine domains. Our study bridges the gap between general-purpose annotation research and the more rigorous demands of domain-specific annotation tasks, offering a critical assessment of whether LLMs can serve as direct, out-of-the-box alternatives to expert annotators.

# 6    Conclusion

In this work, we investigate the research question: "Can top-performing LLMs – often perceived as having expert-level proficiency on academic and professional benchmarks – serve as direct alternatives to human expert annotators?" Our findings suggest that LLMs equipped with inference-time techniques (*e.g.*, CoT, self-consistency) yield only marginal, and sometimes even negative, performance gains. While reasoning models may perform slightly better, they do not show statistically significant improvements in most settings, and still fall short of human expert performance. In addition, we introduce a multi-agent discussion framework that effectively enhances accuracy and consensus, outperforming individual LLMs. Finally, we conduct a fine-grained analysis and identify specific model behaviors emerging during the multi-agent discussion process.

Our results – spanning both single-LLM and multi-agent methods across three domains and five datasets – indicates that performant LLMs may *not* serve as a direct alternative for annotation tasks requiring domain expertise. Relying solely on parametric knowledge to perform domain-specific, expert-level annotation remains a non-trivial challenge. Given that these specialized domains often involve high-risk sectors, ensuring the precision and accuracy of annotated data is critical. Overall, our findings highlight the gap between existing LLMs and human experts, underscoring the need for future efforts to develop reliable LLMs-as-Expert-Annotators systems.

# Limitation and Future Works

As we aim to provide direct insight and observation on whether top-performing LLMs can perform as expert annotators *out-of-the-box*, we minimize efforts in prompt engineering. Some works have demonstrated that, for specific scenarios, one can achieve sizable improvement through carefully-crafted prompts. Consequently, our results may further benefit from a more exhaustive prompt optimization.

Another potential limitation is that we primarily focus on natural language understanding tasks with fixed label space. Towards a more comprehensive evaluation, natural language generation tasks could be further incorporated. Furthermore, all of our experimental settings involve zero-shot configurations using general-purpose chatbot LLMs. To unveil more of the capabilities of LLMs in annotation tasks, future directions could explore few-shot settings, domain-specific or fine-tuned LLMs tailored to the annotation tasks, retrieval-augmented generation methods, or a promising human-LLM hybrid annotation schema.

## Acknowledgments

This work was supported by National Science and Technology Council, Taiwan, under grants NSTC 113-2634-F-002-003- and 114-2221-E-002-070-MY3, and Ministry of Education (MOE), Taiwan, under grant NTU-114L900901. The work of Chung-Chi Chen was supported in part by AIST policy-based budget project "R&D on Generative AI Foundation Models for the Physical Domain."

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

## A    Datasets

### A.1    Finance

**REFinD** (Kaur et al., 2023) is the largest relation extraction dataset over financial documents, comprising 8 entity pairs and 22 relations, with labels reviewed by financial experts. In this task, annotators are tasked to select the relation type between finance-specific entity pairs, such as [*person*] is an employee of [*organization*].

**FOMC** (Shah et al., 2023) is constructed for identifying sentiments about the future monetary policy stances, annotated by experts with a correlated financial knowledge. The labels of this annotation task are Dovish, Hawkish, and Neutral, where a Dovish sentence indicates easing and a Hawkish sentence indicates tightening.

### A.2    Law

**CUAD** (Contract Understanding Atticus Dataset; Hendrycks et al., 2021) consists of legal contracts with extensive annotations from legal experts, created with a year-long effort by dozens of law student annotators, lawyers, and machine learning researchers. The annotation task is to label 41 types out of legal clauses, classified into 5 answer categories, that are considered important in contract review related to corporate transactions. We manually use "Yes/No" answer category to construct our annotation task as the identification of 32 types of clauses.

**FoDS** (Function of Decision Section dataset; Guha et al., 2024) comprises one-paragraph excerpts from legal decisions, annotated by legal professionals. In this task, annotators are tasked to review a legal decision and identify one out of seven function categories that each section (*i.e.*, excerpt) of the decision serves.

### A.3    Biomedicine

**CODA-19** (COVID-19 Research Aspect Dataset; Huang et al., 2020) codes each segment aspect of English abstracts in the COVID-19 Open Research Dataset (Wang et al., 2020). In this task, annotators are tasked to label each segment as *background*, *purpose*, *method*, *finding/contribution*, or *other* sections. To ensure the quality of the labels, we only adopt instances annotated by biomedical experts. We provide data statistics of the five existing specialized datasets in Table 2.

## B    Annotation Guidelines

We provide annotation guidelines of each dataset from Figure 5 to Figure 11.

## C    Prompt Templates

We provide prompt templates of each methods from Figure 12 to Figure 18.

## D    Additional Results

We provide the accuracy and corresponding p-value of the comparison between reasoning models and non-reasoning models in Table 3 and Table 4.

| Domain | Dataset | Instance Type | #Instances | #Labels |
|---|---|---|---|---|
| Finance | REFinD (Kaur et al., 2023) | Sentence | 200 | 22 |
| | FOMC (Shah et al., 2023) | Sentence | 200 | 3 |
| Law | CUAD (Hendrycks et al., 2021) | Clause | 200 | 32 |
| | FoDS (Guha et al., 2024) | Excerpt | 200 | 7 |
| Biomedicine | CODA-19 (Huang et al., 2020) | Paper Abstract | 200 | 5 |

Table 2: The statistics of existing domain-specific datasets we used in this paper.

| | Finance | | Law | | Biomedicine |
|---|---|---|---|---|---|
| **Accuracy** | **REFinD** | **FOMC** | **CUAD** | **FoDS** | **CODA-19** |
| Best Vanilla | 67.5 | 71.5 | 86.5 | **47.0** | 79.5 |
| Best CoT | 67.0 | 69.5 | 84.0 | 45.5 | 77.5 |
| o3-mini (medium) | 71.5 | 73.0 | **87.0** | 46.5 | 82.0 |
| Claude 3.7 Sonnet (thinking) | **73.0** | **74.0** | 86.5 | 46.5 | **84.5** |

Table 3: Accuracy comparison between reasoning models and non-reasoning models. Text in **bold** indicates the highest accuracy for each dataset.

| | Finance | | Law | | Biomedicine |
|---|---|---|---|---|---|
| **P-value** | **REFinD** | **FOMC** | **CUAD** | **FoDS** | **CODA-19** |
| o3-mini vs. Best vanilla | 0.230 | 0.728 | 1.000 | 1.000 | 0.533 |
| o3-mini vs. Best CoT | 0.093 | 0.249 | 0.263 | 0.856 | 0.150 |
| Claude 3.7 Sonnet vs. Best vanilla | 0.135 | 0.500 | 1.000 | 1.000 | 0.110 |
| Claude 3.7 Sonnet vs. Best CoT | **0.043** | 0.163 | 0.332 | 0.839 | **0.013** |

Table 4: P-values of the significance tests between reasoning models and non-reasoning models. P-values are rounded to three decimal places. Text in **bold** indicates the reasoning model is statistically significantly better than the best non-reasoning models, with p-value $< 0.05$.

Relation Extraction (RE) is the task of extracting relationships between entities in a sentence.

You will be given a sentence that contains two entities: entity 1 and entity 2.
Entity 1 is enclosed in double asterisks (i.e., **entity**) and entity 2 is enclosed in double underscores (i.e., __entity__).
Each entity has its own entity type specified in square brackets before the entity (e.g., [PERSON]**entity 1**).
The definition of the entity types are as follows:
- PERSON: People, including fictional.
- ORG: Companies, agencies, institutions, etc.
- UNIV: Universities, colleges, etc.
- GOV_AGY: Government agencies and departments.
- DATE: Absolute or relative dates or periods.
- GPE: Countries, cities, states.
- MONEY: Monetary values, including unit.
- TITLE: Positions or titles, including military.

Please annotate the relation between entity 1 and entity 2 described in the given sentence according to the following label descriptions.
Note that the relation is directional, meaning that the order of entity 1 and entity 2 matters.
Note that you can only select the most appropriate label that is consist of the given type of entities.
If you think there is no relation or other relation between entity 1 and entity 2, please select the label 0.

- 0: **entity 1** has no relation or other relation to __entity 2__
- 1: [PERSON]**entity 1** has/had the job title of [TITLE]__entity 2__
- 2: [PERSON]**entity 1** is/was an employee of [ORG]__entity 2__
- 3: [PERSON]**entity 1** is/was a member of [ORG]__entity 2__
- 4: [PERSON]**entity 1** is/was a founder of [ORG]__entity 2__
- 5: [PERSON]**entity 1** is/was a employee of [UNIV]__entity 2__
- 6: [PERSON]**entity 1** is/was a member of [UNIV]__entity 2__
- 7: [PERSON]**entity 1** has/had attended [UNIV]__entity 2__
- 8: [PERSON]**entity 1** is/was a member of [GOV_AGY]__entity 2__
- 9: [ORG]**entity 1** is/was formed on [DATE]__entity 2__
- 10: [ORG]**entity 1** is/was acquired on [DATE]__entity 2__
- 11: [ORG]**entity 1** is/was headquartered in [GPE]__entity 2__
- 12: [ORG]**entity 1** has/had operations in [GPE]__entity 2__
- 13: [ORG]**entity 1** is/was formed in [GPE]__entity 2__
- 14: [ORG]**entity 1** has/had shares of [ORG]__entity 2__
- 15: [ORG]**entity 1** is/was a subsidiary of [ORG]__entity 2__
- 16: [ORG]**entity 1** is/was acquired by [ORG]__entity 2__
- 17: [ORG]**entity 1** has/had a agreement with [ORG]__entity 2__
- 18: [ORG]**entity 1** has/had a revenue of [MONEY]__entity 2__
- 19: [ORG]**entity 1** has/had a profit of [MONEY]__entity 2__
- 20: [ORG]**entity 1** has/had a loss of [MONEY]__entity 2__
- 21: [ORG]**entity 1** has/had a cost of [MONEY]__entity 2__

Figure 5: The annotation guideline of REFinD dataset.

Hawkish-Dovish classification is to classify the sentiment about the future monetary policy stance into Dovish, Hawkish, or Neutral.
In general:
- 0: Dovish sentences were any sentence that indicates future monetary policy easing.
- 1: Hawkish sentences were any sentence that would indicate a future monetary policy tightening.
- 2: Neutral sentences were those with mixed sentiment, indicating no change in the monetary policy, or those that were not directly related to monetary policy stance.

You will be given a sentence that falls into one of the following eight categories enclosed in square brackets. Please annotate the sentiment of the sentence according to the following detailed label descriptions.
Note that you can only select one label that is most appropriate.

Detailed label descriptions:
[Economic Status: A sentence pertaining to the state of the economy, relating to unemployment and inflation.]
- 0: when inflation decreases, when unemployment increases, when economic growth is projected as low.
- 1: when inflation increases, when unemployment decreases when economic growth is projected high when economic output is higher than potential supply/actual output when economic slack falls.
- 2: when unemployment rate or growth is unchanged, maintained, or sustained.
[Dollar Value Change: A sentence pertaining to changes such as appreciation or depreciation of value of the United States Dollar on the Foreign Exchange Market.]
- 0: when the dollar appreciates.
- 1: when the dollar depreciates.
- 2: N/A
[Energy/House Prices: A sentence pertaining to changes in prices of real estate, energy commodities, or energy sector as a whole.]
- 0: when oil/energy prices decrease, when house prices decrease.
- 1: when oil/energy prices increase, when house prices increase.
- 2: N/A
[Foreign Nations: A sentence pertaining to trade relations between the United States and a foreign country. If not discussing United States we label neutral.]
- 0: when the US trade deficit decreases.
- 1: when the US trade deficit increases.
- 2: when relating to a foreign nation's economic or trade policy.
[Fed Expectations/Actions/Assets: A sentence that discusses changes in the Fed yields, bond value, reserves, or any other financial asset value.]
- 0: Fed expects subpar inflation, Fed expecting disinflation, narrowing spreads of treasury bonds, decreases in treasury security yields, and reduction of bank reserves.
- 1: Fed expects high inflation, widening spreads of treasury bonds, increase in treasury security yields, increase in TIPS value, increase bank reserves.
- 2: N/A
[Money Supply: A sentence that overtly discusses impact to the money supply or changes in demand.]
- 0: money supply is low, M2 increases, increased demand for loans.
- 1: money supply is high, increased demand for goods, low demand for loans.
- 2: N/A
[Key Words/Phrases: A sentence that contains key word or phrase that would classify it squarely into one of the three label classes, based upon its frequent usage and meaning among particular label classes.]
- 0: when the stance is "accommodative", indicating a focus on "maximum employment" and "price stability".
- 1: indicating a focus on "price stability" and "sustained growth".
- 2: use of phrases "mixed", "moderate", "reaffirmed".
[Labor: A sentence that relates to changes in labor productivity.]
- 0: when productivity increases.
- 1: when productivity decreases.
- 2: N/A

Figure 6: The annotation guideline of FOMC dataset.

A/P Relation classification is to classify the relation between Assessment and Plan Subsection in daily progress notes into DIRECT, INDIRECT, NEITHER, or NOT RELEVANT.

You will be given a pair of passages, Assessment and Plan Subsection, from daily progress notes.
Assessment describes the patient and establishes the main symptoms or problems for their encounter.
Plan Subsection addresses each differential diagnosis/problem with an action plan or treatment plan for the day.

Please annotate the relation between Assessment and Plan Subsection in the given pair according to the following label descriptions.
Note that you can only select one label that is most appropriate.

Label descriptions:
- 0: DIRECT. Assessment section includes a primary diagnosis/problem and it is mentioned in the Plan subsection, or Progress note includes a primary diagnosis/problem for hospitalization and it is mentioned in the Plan subsection, or Plan subsection contains a problem/diagnosis related to the primary signs/symptoms in the Assessment section.
- 1: INDIRECT. Plan subsection contains complications/subsequent events or organ failure related to the primary diagnosis/problem from the Assessment section, or Plan subsection contains other listed diagnoses/problems from the overall Progress Note or in the Assessment section that are not part of the primary diagnosis/problem, or Plan subsection contains a diagnosis/problem that is not previously mentioned but closely related (i.e., same organ system) to the primary diagnoses/problems mentioned in the overall Progress Note or Assessment section.
- 2: NEITHER. None of the criteria for Directly Related or Indirectly Related are met but a diagnosis/problem or other signs/symptoms are mentioned.
- 3: NOT RELEVANT. Plan subsection does not include a diagnosis/problems OR signs/symptoms.

Figure 7: The annotation guideline of AP-Relation dataset.

You will be given one paper abstract comprising several segments.
Each segment is a short text describing a specific aspect of the paper, including background, purpose, method, finding/contribution, or other.

Please annotate the aspects of each segment according to the following label descriptions.
Note that you can only select one label that is most appropriate for each segment. The total number of labels must be equal to the number of segments in the abstract.

Label descriptions:
- 0: Background. "Background" text segments answer one or more of these questions: Why is this problem important?, What relevant works have been created before?, What is still missing in the previous works?, What are the high-level research questions?, How might this help other research or researchers?
- 1: Purpose. "Purpose" text segments answer one or more of these questions: What specific things do the researchers want to do?, What specific knowledge do the researchers want to gain?, What specific hypothesis do the researchers want to test?
- 2: Method. "Method" text segments answer one or more of these questions: How did the researchers do the work or find what they sought?, What are the procedures and steps of the research?
- 3: Finding/Contribution. "Finding/Contribution" text segments answer one or more of these questions: What did the researchers find out?, Did the proposed methods work? Did the thing behave as the researchers expected?
- 4: Other. Text segments that do not fit into any of the four categories above. Text segments that are not part of the article. Text segments that are not in English. Text segments that contain only reference marks (e.g., "[1,2,3,4,5]") or dates (e.g., "April 20, 2008"). Captions for figures and tables (e.g. "Figure 1: Experimental Result of ..."). Formatting errors. Text segments the annotator does not know or is not sure about.

Figure 8: The annotation guideline of CODA-19 dataset.

You will be given a clause from a legal contract. Please annotate the category of the given clause according to the following label descriptions.
Note that you can only select one label for each segment that is most appropriate.

Label descriptions:
- 0: Most Favored Nation. This clause provides that if a third party gets better terms on the licensing or sale of technology/goods/services described in the contract, the buyer of such technology/goods/services under the contract shall be entitled to those better terms.
- 1: Non-Compete. This clause imposes a restriction on the ability of a Party to compete with the other party or operate in a certain geography or business or technology sector.
- 2: Exclusivity. This clause provides for an exclusive dealing commitment between the parties of a contract. This clause also includes: a commitment by a party to procure all "requirements" from the other party of certain technology, goods, or services; or a prohibition against licensing or selling technology, goods or services to third parties, or a prohibition on collaborating or working with other parties.
- 3: No-Solicit of Customers. This clause restricts a party from soliciting, contacting or doing business with the other party's customers, vendors or partners.
- 4: Competitive Restriction Exception. This clause states the exception(s) to one of the following three labels: Exclusivity, Non-Compete, or No-Solicit of Customers.
- 5: No-Solicit of Employees. A No-Solicit of Employee clause prohibits a party from soliciting or hiring the other party's employees or consultants for itself or for a third party, during the contract or after the contract ends (or both).
- 6: Non-Disparagement. This clause requires a party not to disparage or defame the other party's goodwill, reputation or image.
- 7: Termination for Convenience. This clause allows a party to terminate a contract without cause or penalty. It allows a party to unilaterally terminate a contract by giving notice and oftentimes after a waiting period expires.
- 8: Right of First Refusal, Offer or Negotiation (Rofr/Rofo/Rofn). This clause grants one party a right of first refusal, right of first offer or right of first negotiation to purchase, license, market, or distribute equity interest, technology, assets, products or services.
- 9: Change of Control. This clause requires consent or notice of the other party if a party undergoes a change of control, such as a merger, stock sale, transfer of all or substantially all of its assets or business (collectively, "CIC").
- 10: Anti-Assignment. This clause requires a party to seek consent or notice if the contract is assigned, transferred or sublicensed to a third party, in whole or in part.
- 11: Revenue/Profit Sharing. This clause requires one party to share revenue or profit with the other party for any technology, goods, or services.
- 12: Price Restriction. This clause restricts the ability of a party to raise or reduce prices of technology, goods, or services provided.
- 13: Minimum Commitment. This clause requires a minimum order size or minimum amount or units per-time period that one party must buy from the counterparty under the contract.
- 14: Volume Restriction. This clause charges a fee or requires consent if one party's use of the product/services exceeds a certain threshold.
- 15: IP Ownership Assignment. This clause provides that intellectual property created by one party becomes the property of the other party, either per the terms of  the contract or upon the occurrence of certain events.

Figure 9: The annotation guideline of CUAD dataset (1-1).

- 16: Joint IP Ownership. This clause provides for joint or shared ownership of intellectual property between the parties to the contract.
- 17: License Grant. This clause authorizes a party to use intellectual property or intangibles of the other party. It can be an authorization to use or to reproduce, distribute, manufacture, etc. certain content, technology, or other items that are protected by intellectual property rights. This clause is very common, and is considered one of the "factual" clauses. The purpose of this label is to help human reviewers to understand what IP is licensed under a contract and what restrictions are imposed on the license, including restrictions on duration, territory and purpose of use.
- 18: Non-Transferable License. This clause prohibits one party to transfer, assign or sublicense IP in the contract.
- 19: Affiliate IP License-Licensor. This clause contains a license grant by affiliates of the licensor or that includes intellectual property of affiliates of the licensor.
- 20: Affiliate IP License-Licensee. This clause contains a license grant to a licensee (incl. sublicensor) and the affiliates of such licensee/sublicensor.
- 21: Unlimited/All-You-Can-Eat License. This clause contains a provision granting one party an "enterprise," "all you can eat" or unlimited usage license.
- 22: Irrevocable or Perpetual License. This clause contains an irrevocable and/or perpetual license of IP. An irrevocable license is a perpetual license that cannot be cut short or terminated. A perpetual license, on the other hand, may not be irrevocable. Namely, a perpetual license can be terminated upon specified events such as material breach. Many license grant clauses use "irrevocable" and "perpetual" in the same sentence. The intent of some contracts may be to use the two terms interchangeably. As a result, for the purpose of CUAD, you should label the two types of licenses under the same label.
- 23: Source Code Escrow. This clause requires one party to deposit its source code into escrow with a third party or into a deposit account with the other party, which can be released to the other party upon the occurrence of certain events (bankruptcy, insolvency, etc.).
- 24: Post-Termination Services. This clause imposes obligations on a party after the termination or expiration of a contract, including any post-termination transition, payment, transfer of IP, wind-down, last-buy, or similar commitments.
- 25: Audit Rights. This clause grants one party the right to audit the books, records, or physical locations of the other party to ensure compliance with the terms of a contract.
- 26: Uncapped Liability. This clause leaves a party's liability uncapped upon the breach of its obligation in the contract. This also includes uncap liability for a particular type of breach such as IP infringement or breach of confidentiality obligation.
- 27: Cap On Liability. This clause includes a cap on liability upon the breach of a party's obligation. This includes time limitation for the counterparty to bring claims or maximum amount for recovery.
- 28: Liquidated Damages. This clause is an agreement to pay a party a pre-determined amount of damages if the other party breaches the contract. For the purpose of CUAD, this clause also includes an early termination fee.
- 29: Insurance. This clause requires a party to maintain insurance for the benefit of the other party.
- 30: Covenant not to Sue. This clause restricts a party from contesting the validity of the other party's ownership of intellectual property or otherwise bringing a claim against the other party that goes beyond the scope of standard Limitation on Liability clauses.
- 31: Third Party Beneficiary. This clause provides that a non-contracting party is a beneficiary to some or all of the clauses in the contract and therefore can enforce its rights against a contracting party.

Figure 10: The annotation guideline of CUAD dataset (1-2).

You will be given a one-paragraph excerpt of a legal decision. Please annotate the category of the given excerpt according to the following label descriptions.
Note that you can only select one label that is most appropriate for the excerpt.

Label descriptions:
- 0: Facts. A section of the decision that recounts the historical events and interactions between the parties that gave rise to the dispute.
- 1: Procedural History. A section of the decision that describes the parties' prior legal filings and prior court decisions that led up to the issue to be resolved by the decision.
- 2: Issue. A section of the decision that describes a legal or factual issue to be considered by the court.
- 3: Rule. A section of the decision that states a legal rule relevant to resolution of the case.
- 4: Analysis. A section of the decision that evaluates an issue before the court by applying governing legal principles to the facts of the case
- 5: Conclusion. A section of the decision that articulates the court's conclusion regarding a question presented to it.
- 6: Decree. A section of the decision that announces and effectuates the court's resolution of the parties' dispute, for example, granting or denying a party's motion or affirming, vacating, reversing, or remanding a lower court's decision.

Figure 11: The annotation guideline of FoDS dataset.

You are a [domain] expert tasked to annotate a [domain] dataset. Please follow the annotation guideline below.

Annotation Guideline:
{[guideline]}

{[instance_type]}:
{[instance]}

Please strictly follow the guideline and output the label in the format of: 'The label is ...'. Do not include any reasoning or explanation.

Figure 12: Vanilla prompt template.

You are a [domain] expert tasked to annotate a [domain] dataset. Please follow the annotation guideline below.

Annotation Guideline:
{[guideline]}

{[instance_type]}:
{[instance]}

Please strictly follow the guideline and output the reasoning and the label in the format of: '**Let's think step by step.** ... The label is ...'.

Figure 13: Chain-of-Thought prompt template.

You are a [domain] expert tasked to annotate a [domain] dataset. Please follow the annotation guideline below.

Annotation Guideline:
{[guideline]}

{[instance_type]}:
{[instance]}

Please strictly follow the guideline and output the reasoning and the label in the format of: '**Let's think step by step.** ... The label is ...'.

Figure 14: Self-Refine prompt template. Step 1: Generate.

You are a [domain] expert tasked to annotate a [domain] dataset. Please follow the annotation guideline below.

Annotation Guideline:
{[guideline]}

{[instance_type]}:
{[instance]}

Please strictly follow the guideline and output the reasoning and the label in the format of: 'Let's think step by step. ... The label is ...'.

**{[model response from step 1.]}**

Review your previous reasoning and annotation and find potential problems. For example, whether the annotation guideline is violated, whether the reasoning is not conclusive.

Figure 15: Self-Refine prompt template. Step 2: Review.

You are a [domain] expert tasked to annotate a [domain] dataset. Please follow the annotation guideline below.

Annotation Guideline:
{[guideline]}

{[instance_type]}:
{[instance]}

Please strictly follow the guideline and output the reasoning and the label in the format of: 'Let's think step by step. ... The label is ...'.

**{[model response from step 1.]}**

Review your previous reasoning and annotation and find potential problems. For example, whether the annotation guideline is violated, whether the reasoning is not conclusive.

Review:
**{[model response from step 2.]}**

Based on the problems you found in the above review, improve your annotation quality and reasoning and output in the format of: 'Let's think step by step ..... The label is ...'.

Figure 16: Self-Refine prompt template. Step 3: Refine.

You are a [domain] expert tasked to annotate a [domain] dataset. Please follow the annotation guideline below.

Annotation Guideline:
{[guideline]}

{[instance_type]}:
{[instance]}

Please strictly follow the guideline and output the reasoning and the label in the format of: 'Let's think step by step. ... The label is ...'.

Figure 17: Multi-agent peer-discussion prompt template. Step 1: Generate initial annotation.

You are a [domain] expert tasked to annotate a [domain] dataset. Please follow the annotation guideline below.

Annotation Guideline:
{[guideline]}

{[instance_type]}:
{[instance]}

Please strictly follow the guideline and output the reasoning and the label in the format of: 'Let's think step by step. ... The label is ...'.

Discussion History:
{Round 1:

Annotator A thinks [model A's output, including reasoning path and the annotated label]
Annotator B thinks [model B's output, ...]
Annotator C thinks [model C's output, ...]

Round 2:
...}

You need to consider the above discussion history carefully. You can maintain your point of view and annotation if others' reasons are not concrete or cannot convince you.
Please strictly follow the guideline and output the reasoning and the label in the format of: 'Let's think step by step. ... The label is ...'.

Figure 18: Multi-agent peer-discussion prompt template. Step 2: Discuss and re-annotate.

