# OpenReview forum: "Evaluating Large Language Models as Expert Annotators"
_colmweb.org/COLM/2025/Conference — COLM 2025_

### Official Review · Reviewer_6CbB · 2025-04-15

**Rating:** 7
**Confidence:** 3
**Ethics Flag:** 1

**Summary:**

**Summary**: This paper presents an extensive evaluation for using LLMs in expert annotation tasks. The evaluation is conducted on five datasets covering three domains that require expert knowledge (biomedicine, law, and finance). The authors first evaluate four state-of-the-art LLMs with various recent prompting techniques and find that none of them lead to a clear improvement across all domains. They then evaluate different combinations of models in a multi-agent discussion framework and find that this does improve the overall performance. The analysis reveals some interesting model behaviors and establishes the correctness of the initial predictions as the upper bound for the multi-agent discussion framework.

* *Quality*: This paper presents a good analysis on using LLMs as expert annotators across different domains, models, and methods. The experiments are well motivated and conducted in a systematic manner (e.g., the authors explicitly focus on datasets that include annotation guidelines).

* *Clarity*: The paper is well written and easy to follow. Some parts could be explained in more detail (see reasons to reject/ questions).

* *Originality*: The paper is more a systematic study of existing methods and models; which somewhat impacts the originality. However, the selection of domains and datasets is somewhat creative and original.

* *Significance*: Considering the recent trend of utilizing LLMs for annotation tasks and for synthetic data generation, the results of this work provide some interesting evidence towards how questionable such usage is. Moreover, this work makes a good case towards a more thorough evaluation of many frequently used approaches.

----

Rebuttal update: raised score to 7

**Questions To Authors:**

Thank you for this interesting work! Although the presented results were quite insightful, there are some open questions where responses would really help to fine-tune the rating:

1. How was the discussion history compiled? Is this just a concatenation of the individual model outputs? Or was there some additional summarization step?
2. Figure 4: Looks quite nice, but could you please elaborate and provide some explicit numbers? What can be observed there?
3. Are you planning to release the full responses of the models for further analysis?

**Reasons To Accept:**

The paper presents a systematic study on the use of LLMs for expert annotation tasks. The paper is well written and clearly motivated; and the conducted experiments cover recent models as well as methods. The results clearly show the limitations of existing methods as well as models and raise some interesting questions for future research.

**Reasons To Reject:**

While it nice that this work presents extensive results on many state-of-the-art models, this is at the same time one of the weaknesses of this work. The use of proprietary models makes it impossible to conduct in-depth analysis which could have been also very insightful (e.g., some attribution analysis to see which parts of the discussion contributed to the final vote). Finally, some details about the conducted experiments is missing which impacts reproducibility (see questions).

---

> ### Author Response · Authors · 2025-06-03
> **Response to Reviewer 6CbB (Part 1)**
>
> Thanks for your detailed feedback!
>
> > *Q1.* How was the discussion history compiled? Is this just a concatenation of the individual model outputs? Or was there some additional summarization step?
> >
>
> The discussion history is compiled via a direct concatenation of the individual model outputs, without any additional summarization. Specifically, each round includes the full reasoning and the annotation from each model, presented in the following format:
>
> `Round 1:`
>
> `Annotator A thinks {model A's output, including reasoning path and the annotated label}`
>
> `Annotator B thinks {model B's output, ...}`
>
> `Annotator C thinks {model C's output, ...}`
>
> `Round 2: ...`
>
> `...`
>
> ---
>
> > *Q2.* Figure 4: Looks quite nice, but could you please elaborate and provide some explicit numbers? What can be observed there?
> >
>
> Here we analyze model behaviors, specifically how models refine or retain their annotations, across two discussion rounds (R1 and R2). Since the observed model behaviors are consistent across datasets, we use FoDS dataset as a representative example. Below we provide detailed explanations along with explicit numbers in Figure 4.
>
> - The 10 darker-colored nodes represent whether the current annotation is correct or not at different stages: darker red (Initial **✔︎**, R1 **✔︎**, R2 **✔︎**), darker green (Initial **✘**, R1 **✘**, R2 **✘**), and darker blue (Consensus **✔︎** / **✘**, Majority Vote (MV) **✔︎** / **✘**).
> - The 3 lighter background colors indicate how models changed or retained their annotation during discussion: light red (initial annotation is correct), light green (initial annotation is wrong), and light blue (all three models reach a consensus).
>
> Note that the number of instances shown in blue (both darker and lighter shades) are the same across three models (these are also marked in **bold** in the corresponding table).

---

> ### Author Response · Authors · 2025-06-03
> **Response to Reviewer 6CbB (Part 2)**
>
> > *Q2.* Figure 4: Looks quite nice, but could you please elaborate and provide some explicit numbers? What can be observed there? (cont'd)
> >
>
> The following tables can be viewed as adjacency matrices, where a directed edge “row node → column node” corresponds to a lighter-colored background region in the figure. The format `number A / number B` indicates that the initial annotation is correct (light red) and incorrect (light green), respectively.
>
> From both the figure and the table, one notable observation is that the reasoning model (Claude 3.7 Sonnet w/ thinking) rarely changes its initial annotations, regardless of correctness (see the upper-right and lower-left `A / B` entries in Claude 3.7’s matrix). While maintaining correct annotations is desirable, this strong self-consistency on incorrect labels potentially limits the benefit of collaborative refinement. In contrast, Gemini 2.0 Flash and GPT-4o are more responsive to others’ reasoning and more willing to revise their annotations. However, they do not consistently identify only the incorrect annotations that need correction.
>
> These behavioral tendencies help explain why our multi-agent framework improves upon individual models, but still falls short of its upper-bound performance -- as collaborative improvement is constrained by both self-consistency tendencies and misidentification of correct answers. We will provide the concrete numbers in Figure 4 and incorporate the above discussion in our revision.
>
> |  | *Claude 3.7* |  |  | **✔︎** |  |  | **\|** |  |  | **✘** |  |  |
> | --- | --- | --- | --- | --- | --- | --- | --- | --- | --- | --- | --- | --- |
> |  |  | Initial | R1 | R2 | MV | Consensus | **\|** | Initial  | R1 | R2 | MV | Consensus |
> |  | Initial | 93 | `21 / -`  |  |  | **72** | **\|** |  | `- / -`  |  |  |  |
> |  | R1 |  | 26 | `8 / 3`  |  | **15** | **\|** |  |  | `- / -` |  |  |
> | **✔︎** | R2 |  |  | 11 | `2 / 2` | **7** | **\|** |  |  |  | `- / -` |  |
> |  | MV |  |  |  | **4** |  | **\|** |  |  |  |  |  |
> |  | Consensus |  |  |  |  | **94** | **\|** |  |  |  |  |  |
> |  |  |  |  |  |  |  |  |  |  |  |  |  |
> |  | Initial |  | `- / 5`  |  |  |  | **\|** | 107 | `- / 38` |  |  | **64** |
> |  | R1 |  |  | `- / -` |  |  | **\|** |  | 38 | `- / 9` |  | **29** |
> | **✘** | R2 |  |  |  | `- / -` |  | **\|** |  |  | 9 | `0 / 3` | **6** |
> |  | MV |  |  |  |  |  | **\|** |  |  |  | **3** |  |
> |  | Consensus |  |  |  |  |  | **\|** |  |  |  |  | **99** |
>
> |  | *Gemini 2.0* |  |  | **✔︎** |  |  | **\|** |  |  | **✘** |  |  |
> | --- | --- | --- | --- | --- | --- | --- | --- | --- | --- | --- | --- | --- |
> |  |  | Initial | R1 | R2 | MV | Consensus | **\|** | Initial  | R1 | R2 | MV | Consensus |
> |  | Initial | 91 | `10 / -` |  |  | **72** | **\|** |  | `9 / -` |  |  |  |
> |  | R1 |  | 22 | `3 / 3` |  | **15** | **\|** |  |  | `- / 1` |  |  |
> | **✔︎** | R2 |  |  | 10 | `2 / 1` | **7** | **\|** |  |  |  | `- / -` |  |
> |  | MV |  |  |  | **4** |  | **\|** |  |  |  |  |  |
> |  | Consensus |  |  |  |  | **94** | **\|** |  |  |  |  |  |
> |  |  |  |  |  |  |  |  |  |  |  |  |  |
> |  | Initial |  | `- / 12` |  |  |  | **\|** | 109 | `- / 33` |  |  | **64** |
> |  | R1 |  |  | `3 / 1` |  |  | **\|** |  | 42 | `1 / 8` |  | **29** |
> | **✘** | R2 |  |  |  | `- / 1` |  | **\|** |  |  | 10 | `- / 3` | **6** |
> |  | MV |  |  |  |  |  | **\|** |  |  |  | **3** |  |
> |  | Consensus |  |  |  |  |  | **\|** |  |  |  |  | **99** |
>
> |  | *GPT-4o* |  |  | **✔︎** |  |  | **\|** |  |  | **✘** |  |  |
> | --- | --- | --- | --- | --- | --- | --- | --- | --- | --- | --- | --- | --- |
> |  |  | Initial | R1 | R2 | MV | Consensus | **\|** | Initial  | R1 | R2 | MV | Consensus |
> |  | Initial | 81 | `12 / -` |  |  | **72** | **\|** |  | `4 / -` |  |  |  |
> |  | R1 |  | 22 | `2 / 2` |  | **15** | **\|** |  |  | `3 / -` |  |  |
> | **✔︎** | R2 |  |  | 9 | `1 / -` | **7** | **\|** |  |  |  | `1 / -` |  |
> |  | MV |  |  |  | **4** |  | **\|** |  |  |  |  |  |
> |  | Consensus |  |  |  |  | **94** | **\|** |  |  |  |  |  |
> |  |  |  |  |  |  |  |  |  |  |  |  |  |
> |  | Initial |  | `- / 10` |  |  |  | **\|** | 112 | `- / 38` |  |  | **64** |
> |  | R1 |  |  | `- / 5` |  |  | **\|** |  | 42 | `2 / 6` |  | **29** |
> | **✘** | R2 |  |  |  | `2 / 1` |  | **\|** |  |  | 11 | `- / 2` | **6** |
> |  | MV |  |  |  |  |  | **\|** |  |  |  | **3** |  |
> |  | Consensus |  |  |  |  |  | **\|** |  |  |  |  | **99** |

---

> ### Author Response · Authors · 2025-06-03
> **Response to Reviewer 6CbB (Part 3)**
>
> > *Q3.* Are you planning to release the full responses of the models for further analysis?
> >
>
> Yes! We plan to release all artifacts, including our implementation of the multi-agent discussion framework, as well as the full responses of the models. The resource will be made available via GitHub to support reproducibility and enable further analysis by the community.

---

> > ### Comment · Reviewer_6CbB · 2025-06-04
> > **Response 6CbB**
> >
> > Thank you for the clarifications! Adding these to the paper will certainly improve the overall quality. I'll raise my score accordingly.

---

> > > ### Author Response · Authors · 2025-06-11
> > > **Follow-Up Response to Reviewer 6CbB**
> > >
> > > Thank you for raising your score! We are glad the clarifications were helpful and are committed to incorporating them to further strengthen our paper. We sincerely appreciate your time and effort in reviewing our work!

---

### Official Review · Reviewer_XTfE · 2025-04-18

**Rating:** 4
**Confidence:** 4
**Ethics Flag:** 1

**Summary:**

The paper proposes a general method for solving different classification tasks, showing that a simulation of inter-annotator discussion at inference timeleads to better matching the gold standard labels across tasks.

**Questions To Authors:**

Why is the approach cast as "annotation" and not just as an ensemble-based classification approach? Is there a fundamental difference?

**Reasons To Accept:**

* The proposed method is straightforwards and applicable to any classification task.

* The experimental results show the positive impact of implementing a strategy that emulates a discussion between agents, both in terms of inter-agent agreement and accuracy.

**Reasons To Reject:**

The selection of specific tasks on which the approach is tested may be critical, and it is not discussed. In fact, the definition of the tasks themselves is only found in the appendix. For one thing, different tasks call for different evaluation metrics. While accuracy is technically fit for any classification task, it may not be the best way to describe the performance, e.g. in the case of unbalanced label distribution. Generative models are also known to be sensitive to how they are prompted. Given the vast difference from one task to another, there need to be a strategy to minimize the dependance of the results from the variations in prompting.

---

> ### Author Response · Authors · 2025-06-03
> **Response to Reviewer XTfE**
>
> Thanks for your valuable feedback!
>
> > The selection of specific tasks on which the approach is tested may be critical, and it is not discussed. In fact, the definition of the tasks themselves is only found in the appendix.
> >
>
> We would like to emphasize that, as outlined in the Introduction (lines 51–53) and Section 2.1, we specifically adopt datasets that
>
> - *(i)* provide fully-detailed annotation guidelines and
> - *(ii)* are manually labelled by domain experts.
>
> Notably, very few existing domain-specific datasets meet both criteria, in particular constraint *(i)*. We will highlight these aspects and incorporate the task definitions and dataset statistics (currently in Appendix A) into the main text.
>
> ---
>
> > For one thing, different tasks call for different evaluation metrics. While accuracy is technically fit for any classification task, it may not be the best way to describe the performance, e.g. in the case of unbalanced label distribution.
> >
>
> We agree that evaluation metrics can be task-dependent, and that accuracy alone is not perfect under class imbalance. Our use of accuracy for evaluation follows established prior works [1][2], in the context of assessing whether top-performing LLMs can serve as direct alternatives to human expert annotators. Specifically, accuracy provides a clear and interpretable measure of agreement quantifying the correctness of LLM-generated labels relative to expert annotations. To provide a more comprehensive evaluation, we will also include F1-score in our updated version.
>
>
> [1] Gilardi et al. "ChatGPT outperforms crowd workers for text-annotation tasks." *arXiv preprint arXiv:2303.15056.* (2023).
>
> [2] He et al. "AnnoLLM: Making Large Language Models to Be Better Crowdsourced Annotators." *Proceedings of the 2024 Conference of the North American Chapter of the Association for Computational Linguistics*. 2024.
>
> ---
>
> > Generative models are also known to be sensitive to how they are prompted. Given the vast difference from one task to another, there need to be a strategy to minimize the dependance of the results from the variations in prompting.
> >
>
> To minimize the impact of prompt variation, we employ a uniform prompt template across all models and tasks (detailed in Lines 115–120 and Appendix C). This standardization of prompt phrasing ensures that the only sources of variation in our results are: *(i)* the annotation guideline -- adapted from each dataset's original paper, and *(ii)* the instance to be labeled.
>
> By controlling all other factors, we reduce variability due to prompt formulation. We agree that any work involving prompting may remain sensitive to prompt design, and our goal is to present a principled and reproducible setup that minimizes prompt dependency across diverse tasks.
>
> ---
>
> > Why is the approach cast as "annotation" and not just as an ensemble-based classification approach? Is there a fundamental difference?
> >
>
> The key distinction lies in how the task is contextualized: we provide fully detailed annotation guidelines to instruct the models, explicitly mimicking the human annotation process. These guidelines specify how to reason about the input in a domain-specific manner. While the annotation task can be viewed as a form of classification at a high level, casting it as "annotation" is both intentional and meaningful, as this setup goes beyond standard ensemble-based classification, where models typically operate with minimal context and without explicit guidance.

---

> > ### Comment · Reviewer_XTfE · 2025-06-10
> >
> > Thank you for the response. the proposed modifications and integrations will surely affect the readability of the paper, in particular providing motivation for the design choices. However, the underlying issues remain substantially unaddressed in this version, therefore I will keep my scores.

---

> > > ### Author Response · Authors · 2025-06-11
> > > **Follow-Up Response to Reviewer XTfE**
> > >
> > > Thank you for your response. We are committed to incorporating the feedback and insights from our discussion to further strengthen our contribution. We sincerely appreciate your time and effort in reviewing our work!

---

### Official Review · Reviewer_CsN1 · 2025-05-09

**Rating:** 6
**Confidence:** 4
**Ethics Flag:** 1

**Summary:**

This work focused on whether large language models (LLMs) can replace human expert annotators in specialized domains. The authors evaluated individual LLMs and multi-agent approaches across finance, biomedicine, and law fields. They empirically observed that (1) chain-of-thought techniques yielded minimal/negative gains, (2) reasoning models excelled in domain-specific annotation, and (3) some models remained stubborn about initial annotations despite contradicting evidence.RetryClaude can make mistakes. Please double-check responses.

**Questions To Authors:**

1) Lack in-depth discussion: the study shows inference methods may not consistently help. What are the possible reasons and what is the significance of this observation? Also, what are some possible reasons for reasoning models not changing their annotations? How is this observation helpful for future tasks?

2) Significance testing: Since the performance gain by the reasoning model is marginal, how significant is the performance improvement?

3) Max discussion round is set to 2: The choice of 2 is a bit arbitrary: if model choice is changed in future, this number of rounds may not be optimal. Why not letting models keep discussing until consensus is reached?

4) Possibility of improving current prompts: The authors mentioned the limitation of missing correct annotation in the initial round be alleviated. I wonder if this is the limitation of current prompt. If you design the peer-discussion prompt to inform that the model’s annotation and other models’ discussion could be both incorrect, can the model take inspiration from other models’ thinking process (not final output) and change annotation to correct one in following rounds?

5) Related work covers limited work. There are works studying LLMs’ capability as expert annotators (e.g. Are Expert-Level Language Models Expert-Level Annotators? (Tseng et al., 2024)) Please cite them and discuss your work’s difference from prior works.

**Reasons To Accept:**

1) The work identifies the gap of evaluating LLM’s annotation capability, benchmarked against expert annotation when most prior efforts use crowdworker annotation.

2) Model and benchmark choices are comprehensive and well justified.

3) Propose an effective multi-agent framework and explore its effectiveness.

**Reasons To Reject:**

1) It will be good to discuss the significance of findings on top of describing the results.

2) Lack discussion of robustness: The authors can vary the prompt and average the performance to increase the robustness of the finding.

3) Suggest to add qualitative analysis on the challenging tasks/instances would be an important component for us to understand LLMs’ capability as expert annotators.

---

> ### Author Response · Authors · 2025-06-03
> **Response to Reviewer CsN1 (Part 1)**
>
> Thanks for the thoughtful review!
>
> > *Q1-1.* What are the possible reasons of why inference methods may not consistently help? What is the significance of this observation?
> >
>
> Inference-time methods may fail to consistently enhance performance due to fundamental limitations in models' ability to understand complex domain-specific contexts. Specifically, models might not accurately interpret specialized annotation guidelines and input instances, thereby failing to capitalize on the additional inference-time compute -- or even degrading performance due to misinterpretation.
>
> These results may offer insight into future domain adaptation strategies. Approaches such as retrieval-augmented prompting or expert-in-the-loop refinement could provide more reliable ways to leverage current-generation LLMs in expert annotation settings.
>
> ---
>
> > *Q1-2.* What are some possible reasons for reasoning models not changing their annotations? How is this observation helpful for future tasks?
> >
>
> We hypothesize two potential explanations for this phenomenon:
>
> - **Overconfidence**: Strong reasoning models may exhibit greater confidence in their initial annotations. This overconfidence can hinder their ability to revise prior judgments, even when presented with correct annotations from other models.
> - **Greater persuasiveness**: As suggested by [1][2], prompts that encourage logical reasoning can yield more persuasive arguments. Consequently, reasoning models may produce more logically coherent and plausible outputs that make others more inclined to align with.
>
> This observation highlights the importance of careful design in future multi-agent systems. It remains an open question whether this effect is ultimately beneficial or detrimental. Should the effect generalize and contribute to unintended consequences, it would warrant a closer look into the role of reasoning models in collaborative LLMs-as-expert-annotators systems, with consideration for appropriate safeguards.
>
> [1] Bozdag et al. "Must Read: A Systematic Survey of Computational Persuasion." *arXiv preprint arXiv:2505.07775* (2025).
>
> [2] Durmus et al. "Measuring the persuasiveness of language models, 2024." *URL https://www.anthropic.com/news/measuring-model-persuasiveness*.

---

> ### Author Response · Authors · 2025-06-03
> **Response to Reviewer CsN1 (Part 2)**
>
> > *Q2.* Significance testing: Since the performance gain by the reasoning model is marginal, how significant is the performance improvement?
> >
>
> We adopt McNemar’s Test to evaluate the statistical significance of performance differences between reasoning models and instruction-tuned models. The table below presents the accuracy and corresponding p-values associated with Figure 1. **Bold** text indicates the highest accuracy for each dataset, and `inline code` text highlights p-values below the significance threshold of 0.05, indicating statistical significance.
>
> Statistical significance is observed in only two datasets when comparing Claude 3.7 Sonnet with thinking enabled against the best instruction-tuned models with CoT. This aligns with our discussed in Section 3.3 that reasoning models only slightly outperform instruction-tuned models. Overall, despite an average accuracy gain of approximately 4%, the advantage of reasoning models may not yet be substantial. We will incorporate the statistical significance results in our revision to further support our findings.
>
> | **Dataset** | **Accuracy** |  |  |  | \| | **p-value*** |  |  |  |
> | --- | --- | --- | --- | --- | --- | --- | --- | --- | --- |
> |  | **Best Vanilla** | **Best CoT** | **o3-mini** | **Claude 3.7 Sonnet** | **\|** | **o3 vs. Best vanilla** | **o3 vs. Best CoT** | **Claude 3.7 vs. Best vanilla** | **Claude 3.7 vs. Best CoT** |
> | REFinD | 67.5 | 67.0 | 71.5 | **73.0** | \| | 0.230 | 0.093 | 0.135 | `0.043` |
> | FOMC | 71.5 | 69.5 | 73.0 | **74.0** | \| | 0.728 | 0.249 | 0.500 | 0.163 |
> | CUAD | 86.5 | 84.0 | **87.0** | 86.5 | \| | 1.000 | 0.263 | 1.000 | 0.332 |
> | FoDS | **47.0** | 45.5 | 46.5 | 46.5 | \| | 1.000 | 0.856 | 1.000 | 0.839 |
> | CODA-19 | 79.5 | 77.5 | 82.0 | **84.5** | \| | 0.533 | 0.150 | 0.110 | `0.013` |
>
> *p-values are rounded to three decimal places.
>
> ---
>
> > *Q3.* Max discussion round is set to 2: The choice of 2 is a bit arbitrary: if model choice is changed in future, this number of rounds may not be optimal. Why not letting models keep discussing until consensus is reached?
> >
>
>
> We set the maximum number of discussion rounds to 2 based on empirical observations and practical considerations:
>
> - We find that nearly all instances reach consensus within 2 rounds. In fact, fewer than 10 instances fail to reach agreement by the end of the second round.
> - For the rare cases that do not reach consensus after 2 rounds, we observe a common pattern: all three agents resist changing their annotations in at least one round, resulting in no progress.
>
> Given these observations, setting the discussion round limit to 2 could provide a good balance between effectiveness and computational efficiency. We agree that dynamically adjusting the number of discussion rounds is a promising direction and we will discuss this as future works.

---

> ### Author Response · Authors · 2025-06-03
> **Response to Reviewer CsN1 (Part 3)**
>
> > *Q4.* Possibility of improving current prompts: The authors mentioned the limitation of missing correct annotation in the initial round be alleviated. I wonder if this is the limitation of current prompt. If you design the peer-discussion prompt to inform that the model’s annotation and other models’ discussion could be both incorrect, can the model take inspiration from other models’ thinking process (not final output) and change annotation to correct one in following rounds?
> >
>
> We agree that framing the prompt to explicitly acknowledge the possibility of incorrect context could help guide the model toward better annotations. Our work serves as an important first step, paving the way for such future exploration of alternative annotation system setups and prompt design strategies!
>
> ---
>
> > *Q5.* Related work covers limited work. There are works studying LLMs’ capability as expert annotators (e.g. Are Expert-Level Language Models Expert-Level Annotators? (Tseng et al., 2024)) Please cite them and discuss your work’s difference from prior works.
> >
>
> Our work differs from and significantly extends [3] along multiple dimensions. Specifically:
>
> - We incorporate an experimental setup involving reasoning models both as individual LLMs and within the multi-agent framework, thereby enabling a more comprehensive methodological investigation.
> - We provide an in-depth, fine-grained analysis of the conditions under which the multi-agent discussion framework is effective or limited, offering richer insights for practitioners.
> - We observe and highlight a key behavioral difference in multi-agent LLMs-as-expert-annotators systems: reasoning models tend to maintain their annotations during discussions, in contrast to instruction-tuned models.
>
> We will extend the related work section to include [3] and discuss other relevant works.
>
> [3] Tseng et al. "Are Expert-Level Language Models Expert-Level Annotators?" arXiv preprint arXiv:2410.03254 (2024).
>
> ---
>
> > Suggest to add qualitative analysis on the challenging tasks/instances would be an important component for us to understand LLMs’ capability as expert annotators.
> >
>
> Thank you for the suggestion! We will incorporate additional qualitative analysis in our revision to better highlight models’ capabilities on challenging instances.

---

> > ### Comment · Reviewer_CsN1 · 2025-06-08
> > **Response to authors**
> >
> > Thank you for the detailed response. After careful consideration, I will maintain my scores for this submission. I encourage the authors to incorporate deeper insights and more discussions. For instance (but not limit to), analysis of what causes multi-agent discussion failures, how different prompting engineering strategies affect performance, and the efficiency and complexity of multi-agent vs single model. These insights would strengthen the contribution and provide valuable guidance for future work in this area.

---

> > > ### Author Response · Authors · 2025-06-11
> > > **Follow-Up Response to Reviewer CsN1**
> > >
> > > Thank you for your response. We are committed to incorporating the feedback and insights from our discussion to further strengthen our contribution. We sincerely appreciate your time and effort in reviewing our work!

---

### Official Review · Reviewer_jNni · 2025-05-15

**Rating:** 8
**Confidence:** 4
**Ethics Flag:** 1

**Summary:**

This paper explores the use of large language models (LLMs) as annotators in specialized domains such as finance, law, and biomedicine. The authors evaluate both instruction-tuned and reasoning-focused models on five datasets, using only annotation guidelines and unlabeled data. They also test inference-time techniques like chain-of-thought prompting, self-refinement, and self-consistency. Results show that inference-time techniques offer limited benefit for instruction-tuned models, while reasoning models perform slightly better overall. Multi-agent collaboration improves both accuracy and model agreement, but remains limited when initial predictions lack the correct answer. The analysis also finds that reasoning models are more rigid, while models like Gemini 2.0 Flash and GPT-4o revise outputs more flexibly.

**Questions To Authors:**

1. Why do you think inference-time techniques such as chain-of-thought prompting, self-consistency, and self-refinement underperform in your expert-annotation tasks, despite prior work showing their effectiveness in other NLP tasks? Does the domain-specific nature of the required knowledge plays a role? Could you elaborate on possible reasons?
2. Do you expect the observed behaviors (particularly the failure of inference-time techniques and the success of multi-agent collaboration) to generalize beyond expert domains? Or do you think these are specific to the complexity and implicit knowledge requirements of domains like finance, law, and biomedicine?

**Reasons To Accept:**

- The paper offers a comprehensive evaluation of LLMs as annotators across specialized domains (finance, law, and biomedicine) and addresses an underexplored but practically important area.
- It compares a diverse set of models (i.e., instruction-tuned and reasoning models: Clause-3-Ops, Gemini 1.5 Pro, Gemini 2.0 Flash, GPT-4o, Clause-3.7-Sonnet, o3-mini) and inference-time techniques (i.e., CoT, self-consistency, self-refine, and multi-agent discussion), providing insights into their relative strengths and weaknesses.
- The analysis of multi-agent discussion is detailed and insightful, showing both performance improvements and inherent limitations.
- The paper presents results clearly through informative figures and tables, making the findings easy to interpret.

**Reasons To Reject:**

Overall, this paper is highly interesting. My main concern lies in the limited discussion surrounding the key empirical observations reported in the experiments, specifically, why inference-time techniques underperform while multi-agent collaboration proves effective. These contrasting results are intriguing, yet the paper does not offer sufficient explanation or analysis to help the reader understand the underlying causes.

1. The paper reports that inference-time techniques (e.g., chain-of-thought, self-refine, self-consistency) tend to degrade performance when applied to instruction-tuned models (while multi-agent collaboration proves effective). However, it does not provide a clear explanation for this phenomenon. This is especially notable given that prior work has shown these inference-time techniques to improve performance across a variety of tasks. Since expert-annotation tasks share a similar input-output formalization with those in existing studies, the lack of improvement raises important questions. For example, could the domain-specific nature of the required knowledge limit the effectiveness of these techniques? (If so, why?) Addressing such questions would enhance the paper’s depth and its potential to inspire further discussion within the community.
2. The paper does not discuss whether the findings (such as the limited effectiveness of inference-time techniques and the benefits of multi-agent collaboration) are specific to expert domains (e.g., finance, law, biomedicine) or generalizable to other domains. Clarifying this point would help position the work more clearly within the broader literature.
3. Important practical aspects such as annotation cost, inference latency, and scalability are not discussed, which limits the study’s implications for real-world deployment.

---

> ### Author Response · Authors · 2025-06-03
> **Response to Reviewer jNni**
>
> Thanks for the thoughtful review!
>
> > *Q1.* Why do you think inference-time techniques such as chain-of-thought prompting, self-consistency, and self-refinement underperform in your expert-annotation tasks, despite prior work showing their effectiveness in other NLP tasks? Does the domain-specific nature of the required knowledge plays a role? Could you elaborate on possible reasons?
> >
>
> Inference-time methods may fail to consistently enhance performance due to fundamental limitations in models’ ability to understand complex domain-specific contexts. Specifically, models might not accurately interpret specialized annotation guidelines and input instances, thereby failing to capitalize on the additional inference-time compute -- or even degrading performance due to misinterpretation.
>
> These results may offer insight into future domain adaptation strategies. Approaches such as retrieval-augmented prompting or expert-in-the-loop refinement could provide more reliable ways to leverage current-generation LLMs in expert annotation settings.
>
> ---
>
> > *Q2.* Do you expect the observed behaviors (particularly the failure of inference-time techniques and the success of multi-agent collaboration) to generalize beyond expert domains? Or do you think these are specific to the complexity and implicit knowledge requirements of domains like finance, law, and biomedicine?
> >
>
> We posit that the observed behaviors are more characteristic of specialized domains that demand a complex understanding of domain-specific context. While inference-time techniques have demonstrated effectiveness in broader tasks, we hypothesize that their generalizability diminishes as task complexity and domain expertise requirements increase.
>
> Our empirical results suggest that, in such settings, inference-time strategy often struggle -- potentially due to their limited ability to access or infer specialized knowledge. In contrast, multi-agent collaboration could more fundamentally enhance the system's capacity (and has also been shown to be effective in general domain tasks [1]), enabling models to contribute complementary perspectives, knowledge, and reasoning paths -- capabilities that individual LLMs or single-turn inference methods may not consistently achieve.
>
> [1] Du, Yilun, et al. "Improving factuality and reasoning in language models through multiagent debate." *Forty-first International Conference on Machine Learning*. 2023.

---

### Decision · Program_Chairs · 2025-07-08

**Decision:**

Accept

**Comment:**

I recommend accepting this paper. It provides a thorough evaluation of LLMs as expert annotators across specialized domains, with novel insights about their collaborative behaviors in a multi-agent discussion framework. Three of four reviewers supported acceptance, and the authors addressed key concerns in their rebuttal. The findings about reasoning models being less willing to change their initial annotations, while instruction-tuned models show more flexibility, offer valuable insights for deploying LLMs in expert annotation tasks. This work makes a meaningful contribution to an important practical problem.